# IGF-1 as a Potential Therapy for Spinocerebellar Ataxia Type 3

**DOI:** 10.3390/biomedicines10020505

**Published:** 2022-02-21

**Authors:** Yong-Shiou Lin, Wen-Ling Cheng, Jui-Chih Chang, Ta-Tsung Lin, Yi-Chun Chao, Chin-San Liu

**Affiliations:** 1Vascular and Genomic Center, Institute of ATP, Changhua Christian Hospital, Changhua 50091, Taiwan; 398157@cch.org.tw (Y.-S.L.); 111800@cch.org.tw (W.-L.C.); 98333@cch.org.tw (T.-T.L.); 2Center of Regenerative Medicine and Tissue Repair, Institute of ATP, Changhua Christian Hospital, Changhua 50091, Taiwan; 145520@cch.org.tw; 3General Research Laboratory of Research Department, Changhua Christian Hospital, Changhua 50091, Taiwan; 4Inflammation Research & Drug Development Center, Changhua Christian Hospital, Changhua 50091, Taiwan; 183782@cch.org.tw; 5Department of Neurology, Changhua Christian Hospital, Changhua 50091, Taiwan; 6Graduate Institute of Integrated Medicine College of Chinese Medicine, China Medical University, Taichung 40402, Taiwan; 7Department of Post-Baccalaureate Medicine, College of Medicine, National Chung Hsing University, Taichung 402, Taiwan

**Keywords:** spinocerebellar ataxia type 3, insulin-like growth factor-1, locomotor function, Purkinje cells, mitochondrial function, autophagy

## Abstract

Although the effects of growth hormone (GH) therapy on spinocerebellar ataxia type 3 (SCA3) have been examined in transgenic SCA3 mice, it still poses a nonnegligible risk of cancer when used for a long term. This study investigated the efficacy of IGF-1, a downstream mediator of GH, in vivo for SCA3 treatment. IGF-1 (50 mg/kg) or saline, once a week, was intraperitoneally injected to SCA3 84Q transgenic mice harboring a human ATXN3 gene with a pathogenic expanded 84 cytosine–adenine–guanine (CAG) repeat motif at 9 months of age. Compared with the control mice harboring a 15 CAG repeat motif, the SCA3 84Q mice treated with IGF-1 for 9 months exhibited the improvement only in locomotor function and minimized degeneration of the cerebellar cortex as indicated by the survival of more Purkinje cells with a more favorable mitochondrial function along with a decrease in oxidative stress caused by DNA damage. These findings could be attributable to the inhibition of mitochondrial fission, resulting in mitochondrial fusion, and decreased immunofluorescence staining in aggresome formation and ataxin-3 mutant protein levels, possibly through the enhancement of autophagy. The findings of this study show the therapeutic potential effect of IGF-1 injection for SCA3 to prevent the exacerbation of disease progress.

## 1. Introduction

Spinocerebellar ataxia (SCA) is one of the most common genetic neurodegenerative diseases with multiple types. Among the different types of SCA, SCA type 3 (SCA3), also known as Machado–Joseph disease, is highly expressed in the Asian population and is the common type of autosomal dominant SCA in Taiwan [1]. Compared with patients with SCA1 and SCA2, those with SCA3 have a later onset and present with cerebellar ataxia, peripheral neuropathy, distal muscle atrophy, and hyporeflexia [2]. These symptoms continue to worsen with time, and no known effective treatment is currently available. The SCA3 locus responsible for SCA3 is located on chromosome 14q32.1 [3], and its causative gene ATXN3 causes an abnormal expansion of cytosine–adenine–guanine (CAG) trinucleotide repeats, leading to the production of the toxic protein polyglutamine (polyQ) [4]. PolyQ diseases, including SCA, spinal and bulbar muscular atrophy (SBMA), and Huntington’s disease (HD), are currently identified as a group of neurodegenerative disorders [5]. Because of the abnormal accumulation of polyQ, ataxin-3 loses the function of deubiquitinating enzymes and affects the efficiency of proteasome degradation [6], leading to imbalances in regulation pathways and ultimately neuronal apoptosis [7]. Aggregation and toxicity caused by mutant ataxin-3 are the hallmarks of SCA3 [8]. Therefore, the clearance of mutant ataxin-3 proteins is proposed as a promising therapeutic strategy [9].

Human growth hormone (GH), secreted by the anterior pituitary, is a signaling molecule that affects the CNS, cell proliferation and differentiation, and metabolism [10,11]. Our previous study reported that intraperitoneal (IP) injection of GH restored locomotor functions and preserved more Purkinje cells (PCs)/cerebellar cortex cells in SCA3 84Q transgenic mice [12]. In addition to activating GH receptors, GH induces the liver or cerebral and peripheral nerve cells to synthesize and secrete insulin-like growth factor 1 (IGF-1) [13]. GH and IGF-1 regulate cellular function. Impaired release of GH and IGF-1 can cause substantial changes in tissue structures and functions [14]. Compared with GH, IGF-1 exerts multiple effects on the nervous system and has a stronger effect on motor and sensory neurons and neurogenesis [15,16]. IGF-1 is not only involved in the regulation of obesity, cancer, metabolism, and aging [17], but also affects the survival and maturation of cells [18]. For tissue homeostasis, IGF-1 is an essential paracrine and autocrine factor [19] with antiapoptotic and anti-inflammatory functions [20]. Moreover, IGF-1 has been demonstrated to be beneficial in amyotrophic lateral sclerosis (ALS) [21,22].

Glypromate (glycine–proline–glutamate, GPE) is an N-terminal tripeptide of IGF-1 that exerts a neuroprotective effect, and its structure and biological function are similar to those of IGF-1 [23,24]. GPE can not only easily cross the blood–brain barrier, but also be naturally cleaved by an acid protease in the brain [25]. The efficacy of the daily subcutaneous injection of IGF-1 in patients with SCA3 has been validated [26,27]. However, these were open-label studies, and the effect of IGF-1 should be experimentally verified because these studies included a limited number of patients, had a short follow-up period, and did not elucidate the molecular mechanism of IGF-1. Moreover, the treatment efficacy of IGF-1 when used for more than several months has not been studied in patients with SCA3. In laboratory rodents, IP administration is safer, better tolerated, less stressful, and more suitable for repeated administration as well as has a higher absorption rate [28]. Thus, in this study, the protective effect and possible mechanisms of IGF-1, especially in the regulation of mitochondrial function and autophagy in aggregate protein clearance, the critical factors responsible for IGF-1 benefit in SCA3 treatment, were investigated in SCA3 84Q transgenic mice that were administered a weekly IP injection of IGF-1 for 9 months (the same approach was used for GH intervention in our previous study).

## 2. Materials and Methods

### 2.1. Animal Model

SCA3 15Q transgenic mice were a generous gift from Prof. Henry L. Paulson (Department of Neurology, University of Michigan, Ann Arbor, MI, USA), and SCA3 84Q transgenic mice were purchased from Jackson Laboratory (Bar Harbor, ME, USA). The genome of transgenic mouse strains contains a yeast artificial chromosome transgene that expresses the human ataxin-3 gene modified with expanded 15 and 84 CAG repeat motifs that mimic the health and SCA3 in humans, respectively. All the mice were of C57BL/6J background, and the identities of the SCA transgenic mice were confirmed through the PCR of a DNA sample obtained from the mouse tail. The primers used for the detection of the CAG repeat in the ataxin-3 gene were as follows: forward primer 5′-TGGCCTTTCACATGGATGTGAA-3′ and reverse primer 5′-CCAGTGACTACTTTGATTCG-3′. The 163 bp molecule belongs to SCA3 15Q, whereas that 430 bp molecule refers to SCA3 84Q. All the mice were housed under a 12 h light/dark cycle. All the animal experiments were approved by the Institutional Animal Care and Use Committee of the Changhua Christian Hospital (approval No. CCH-AE-106-017, 2 February 2018). The transgenic mice were divided into three groups: SCA3 15Q mice as the normal control group (n = 6), SCA3 84Q mice treated with saline as the sham control group (n = 8), and SCA3 84Q mice treated with IGF-1 as the study group (n = 8). A low dosage of IGF-1 was used in this study following the method reported by Deacon et al. [29]. Each IGF-1- and saline-treated SCA3 84Q mouse intraperitoneally received 50 mg/kg of IGF-1 (GPE, Sigma-Aldrich, St. Louis, MO, USA) and saline, respectively, weekly from postnatal age of 9 months to 18 months. The body weight of the mice was recorded once a week to monitor their health.

### 2.2. Rotarod Test

Motor coordination and balance were evaluated using a rotarod apparatus (Accelerating Model, Ugo Basile Biological Research Apparatus, Varese, Italy) in the mice once a month during the intervention period. The mice were put on a rod (6 cm in diameter) at a preliminary speed of 5 rpm. The speed of the rod was linearly increased from 5 to 40 rpm within 300 s. The latency to fall (height = 20 cm) was defined as the period (in seconds) for which a mouse persisted on the rod, and the maximal period for each mouse was 5 min. After a mouse was removed from the rod, it was allowed to rest for 20 min before the next trial to prevent exhaustion. Each mouse underwent six trials, and the latency to fall was recorded and analyzed statistically.

### 2.3. Open Field Test

Behavioral experiments were conducted using a carton with opaque bottom (carton size: 45 cm long × 45 cm wide × 65 cm high). An infrared sensitive camera was placed on the top of the carton and operated using the EthoVision XT 7.0 (Noldus Information Technology, Wageningen, The Netherlands) software. This video-tracking software automatically tracked mouse activities, namely the move distance, movement time, walking distribution, and velocity in 10 min. The walking distribution was calculated by dividing the field of view into four quadrants and counting the number of times the mouse crossed the quadrant. Behavioral analysis was performed before the mice were euthanized.

### 2.4. Catwalk Gait Analysis

Gait analysis was performed before the mice were euthanized, and the method used was described previously [30]. Briefly, the gait of the mice was analyzed using the CatWalk automated gait analysis system (Noldus Information Technology, Wageningen, The Netherlands). The apparatus consisted of a long glass plate as the walkway floor for the mouse to traverse voluntarily. Under the glass plate, a high-speed camera was placed. When a mouse traversed from one end to the other, its footsteps were illuminated by fluorescent light emitted from below and recorded by the camera. Each mouse was subjected to 8–10 trials, and the statistics of multiple parameters, namely step cycle (time between two consecutive initial contacts of the same paw), stride length (the distance between successive placements of the same paw), stand (duration of contact of a paw with the glass plate), and average speed, were recorded and analyzed using CatWalk XT 9.0.

### 2.5. Histological Tissue Section Preparation

The mouse hemibrain was harvested at 18 months of age, fixed with 4% paraformaldehyde for 48 h, and then dehydrated. The tissue was soaked in xylene to remove alcohol and fat. Subsequently, the tissue was submerged in molten paraffin in a mold. After the hardening of paraffin, the embedded tissue was removed from the mold. A microtome was used to obtain 5 μm-thick tissue sections, which were then transferred to a glass slide.

### 2.6. Hematoxylin and Eosin Staining

To observe the cerebellar molecular layer (ML), granular layer (GL), and PCs, the tissues were stained with H&E. The sections were deparaffinized and sequentially rehydrated in xylene and 100%, 95%, 80%, and 75% ethanol and water, followed by staining with H&E. Subsequently, the sections were immersed in 80%, 95%, and 100% ethanol and xylene for dehydration and then sealed.

### 2.7. Immunohistochemical Staining of Ataxin-3 and 8-OHdG

After deparaffinization and rehydration, antigen retrieval was performed using an antigen retriever apparatus with heat and pressure. The sections were washed in 1× PBS (10 mM Tris–HCl, 150 mM NaCl, 0.05% (v/v) Tween 20, pH 8.0) and blocked with 3% hydrogen peroxide at room temperature (RT) for 10 min. Subsequently, the sections were incubated with the anti-ataxin-3 or anti-8-OHdG monoclonal antibody diluted in the blocking solution (1:300) for 40 min at RT. After washing with 1× PBS, the sections were incubated with a secondary antibody diluted in the blocking solution (1:1000) for 40 min at RT. After washing, the sections were incubated with peroxidase-conjugated streptavidin for 20 min at RT. To achieve an appropriate level of staining, the sections were developed under a microscope by using diaminobenzidine for approximately 1 min. Then, the sections were counterstained with hematoxylin, dehydrated through sequential immersion in alcohol and xylene, and coverslipped. Ten microscopic fields of posterior lobules were examined in each section and analyzed using the ImageJ software (Rasband, W.S., National Institutes of Health, Bethesda, MD, USA).

### 2.8. Immunofluorescence Staining

Paraffin-embedded tissue sections were deparaffinized before staining. The sections were immersed in xylene twice every 10 min to dissolve paraffin. Before the final wash with water, a series of decreasing alcohol concentrations was used to remove xylene. The sections were then fixed with 4% formaldehyde in PBS at 37 °C for 15 min. The sections were placed in 0.01 M citrate buffer and heated to 95 °C for 40 min to retrieve the antigen. Subsequently, the sections were washed with deionized water and cooled for 15 min. The sections were then stained with the ProteoStat^®^ dye (1000× in PBS from a ProteoStat^®^ Aggresome Detection Kit, Enzo Life Sciences, Farmingdale, NY, USA) for 3 min. The sections were immersed in deionized water and destained in 1% acetic acid for 20 min and then blocked in PBS containing 3% BSA at RT for 30 min. Subsequently, the sections were incubated with anti-ataxin-3 (Abcam, Cambridge, MA, USA, ab175265, 1:50) at 4 °C overnight. After washing with PBS three times, the sections were incubated with goat anti-rabbit secondary antibodies conjugated to DyLight 488 (1:200, Jackson ImmunoResearch, West Grove, PA, USA) and 10 μg/mL Hoechst 33,342 (Invitrogen Molecular Probes, Paisley, UK) for 1 h. The sections were then washed three times with PBS and cooled for 15 min. Finally, the coverslips were mounted on slides by using 50% glycerol. Imaging was performed using a phase-contrast microscope (IX81, Olympus, Tokyo, Japan) with a 400× digital camera (DP72, Olympus). Images were obtained from different parts of the cerebellum and analyzed using the ImageJ software.

### 2.9. Western Blot Analysis

The mouse cerebellum was lysed in the T-PER^TM^ Tissue Protein Extraction Reagent (Thermo Fisher Scientific, Rockford, USA) supplemented with Phosphatase Inhibitor Cocktail Set V (Millipore, Darmstadt, Germany) and Protease Inhibitor Cocktail Set I (Millipore) at 4 °C for 2 h. After centrifugation at 14,000 rpm at 4 °C for 30 min, the supernatant was collected. The protein concentration was measured and quantified using the Pierce^TM^ BCA Protein Assay Kit (Thermo Fisher Scientific). The proteins were denatured in 6× SDS loading dye (AllBio, Taichung, Taiwan) at 95 °C for 5 min. Then, 15–40 μg of proteins from each sample were loaded on 10% or 12% SDS–PAGE gels (Bio-Rad Laboratories, Richmond, CA, USA) and then transferred onto a PVDF membrane (Millipore). The membranes were blocked in a BlockPRO Blocking Buffer (Visual Protein, Taipei, Taiwan) at RT for 2 h and then incubated at 4 °C with the following primary antibodies overnight: Calbindin (Cell Signaling, Danvers, Massachusetts, USA, #2173, 1:1000), Drp1 (Abcam, ab154879, 1:500), p-Drp1 (Cell Signaling, #3455, 1:250), Opa1 (BD Biosciences, San Jose, CA, USA, #612607, 1:500), Mfn2 (Sigma-Aldrich, M6319, 1:500), Ataxin-3 (Abcam, ab175265, 1:500), Beclin 1 (Novusbio, NB110-87318, 1:1000), p62 (Sigma-Aldrich, P0067, 1:1000), Atg7 (Abcam, ab133528, 1:2500), LC3A/B (Cell Signaling, #4108, 1:500), and Lamp2 (Proteintech, Rosemont, USA, 10397-1-AP, 1:500). After washing three times with TBS (0.1% Tween 20), the membrane was incubated with anti-mouse or anti-rabbit secondary antibodies (Jackson Immuno Research Laboratories, Pennsylvania, USA, 1:10,000) at RT for 1 h. Chemiluminescent signals were detected using the Immobilon Western Chemiluminescent HRP Substrate (Millipore) on a Fusion-FX7-826.WL Superbright Transilluminator instrument (Vilber Lourmat, Eberhardzell, Germany). The band intensities were normalized to the total expression of proteins and quantitated using the ImageJ software.

### 2.10. Cerebellar Mitochondrial Functional Measurement

After euthanizing the mice, a part of their cerebellum was ground immediately, and a high-resolution respirometry device (Oxygraph-2k (O2k); Oroboros, Innsbruck, Austria) was used to detect cerebellum mitochondrial respiration data. To examine mitochondrial respiration, 0.5 mM malate and 10 mM L-glutamate were added to obtain complex I-linked LEAK respiration. Subsequently, 2.5 mM ADP was added to induce complex I-linked oxidative phosphorylation (OXPHOS). Then, 10 mM succinate was added to observe CI + II-linked OXPHOS. Subsequent injections of the ATP synthase inhibitor oligomycin (5 μM) and the uncoupler FCCP (1.5 μM) allowed examining the LEAK respiration in the presence of adenylates and the maximum noncoupled respiration, respectively. Finally, the electron transport chain inhibitor rotenone (10 μM) and antimycin A (6.25 μM) were added to inhibit CI and CIII, respectively, to completely shut down mitochondrial oxygen consumption and obtain a measure of residual oxygen consumption (ROX).

### 2.11. Plasma Neurofilament Light Chain Measurement

Blood samples were taken into anticoagulation tubes (BD Microtainer, Becton, Dickinson and Company, Baltimore, USA) and centrifuged at 2500 rpm for 10 min at RT within 2 h of collection. Plasma supernatants were collected and divided into aliquots, then frozen at −80 °C until used for analyses. The plasma level of the neurofilament light chain (Nf-L) was measured using a competitive Nf-L ELISA kit (10-7002, UmanDiagnostics, Umeå, Sweden), in which a monoclonal antibody specific for Nf-L had been precoated onto a microplate according to the manufacturer’s instructions.

### 2.12. Statistical Analysis

All the data were presented as the means ± SEM. Graphics were generated using the GraphPad Prism software (version 7.0, Graph Pad Software, San Diego, CA, USA). The SPSS (SPSS Statistics for Windows, Version 17.0, SPSS Inc., Chicago, CA, USA) or GraphPad software was used for statistical analysis. Statistical analysis was determined by one-way analysis of variance (ANOVA) followed by the Bonferroni multiple comparisons post hoc test or two-way repeated measures ANOVA followed by the Bonferroni multiple comparisons post hoc test (rotarod test). A difference with *p* < 0.05 was considered statistically significant and is indicated with an asterisk.

## 3. Results

### 3.1. IGF-1 Maintained the Motor Ability of SCA3 mice

To examine the motor coordination and gait performance of the transgenic mice with or without treatment for 9 months, we performed rotarod, behavior box, and CatWalk gait tests. The rotarod test was performed once a month to evaluate motor coordination and endurance in the mice. Throughout the experimental process, the saline-treated SCA3 84Q mice displayed a shorter time of latency to fall than did the SCA3 15Q and IGF-1-treated SCA3 84Q mice (Figure 1a). After 9 months of treatment, the average latency to fall was longer in the IGF-1-treated SCA3 84Q mice than in the saline-treated SCA3 84Q mice (128.50 ± 11.95 s vs. 93.06 ± 10.35 s, *p* > 0.05) (Figure 1a), but did not reach statistical significance. After normalization to the pretreatment point, the percentage of relative latency to fall in the IGF-1-treated SCA3 84Q mice was still higher than that in the saline-treated SCA3 84Q mice (93.40% ± 15.31% vs. 55.91% ± 4.07%; *p* > 0.05), but did not reach a significant level (Figure 1b). This trend continued throughout the 9-month treatment.

The behavior box test was conducted at the end of the study when the mice were 18 months old. As shown in Figure 1c, the Etho-Vision XT 7.0 software was used to record and analyze the mouse movement path and the autonomous movement ability. In all the locomotor activities, the performance of the IGF-1-treated SCA3 84Q mice was more favorable than that of the saline-treated SCA3 84Q mice in the ninth month including the distance moved (1903.26 ± 277.98 cm vs. 1113.99 ± 219.42 cm; *p* < 0.05), movement (257.66 ± 28.38 s vs. 156.73 ± 35.67 s), frequency of zone change (59.50 ± 7.80 times vs. 37.33 ± 9.82 times), and velocity (3.71 ± 0.62 cm/s vs. 2.00 ± 0.39 cm/s; *p* < 0.05) (Figure 1d).

In the catwalk gait analysis, mouse footprints were captured and converted into images (Figure 1e). The measurements of the footprints contained the step cycle, stride length, stand, and average speed (Figure 1f) and were analyzed using the software that was linked to the apparatus. For these four parameters, the test values of the saline-treated SCA3 84Q mice significantly differed from those of the SCA3 15Q mice for the leg sets, indicating a change in gait performance owing to SCA3. Although no significant change after treatment was observed in the IGF-1-treated SCA3 84Q mice compared with the saline-treated SCA3 84Q mice, an improved trend in step cycle and stand was still observed in the IGF-1-treated SCA3 84Q mice. This finding indicated that IGF-1 treatment might confer partial protective effects on the SCA3 mice.

### 3.2. IGF-1 Restored the Loss of PCs and the Thickness of GL and ML in The Mouse Cerebellum

After the motor ability tests, the 18-month-old mice were euthanized. Their brains were harvested and used for histocytological analysis, which involved the visualization of the PC layer (PCL), GL, and ML. The number of PCs along the PCL of the posterior lobules of the cerebellum was counted. The average number of PCs per 100 μm in the PCL was quantified. The saline-treated SCA3 84Q mice had a significantly lower number of PCs than did the SCA3 15Q mice (2.20 ± 0.05 vs. 3.04 ± 0.15; *p* < 0.05); the number of PCs was restored to a normal level after the IGF-1 treatment (IGF-1-treated SCA3 84Q vs. saline-treated SCA3 84Q, 2.57 ± 0.06 vs. 2.20 ± 0.05; *p* < 0.05) (Figure 2a). The results of the Western blot revealed that the calbindin level was increased in the cerebellum extracts of the IGF-1-treated SCA3 84Q mice (Figure 2b).

The average thickness of the GL in the saline-treated SCA3 84Q mice was significantly lower (saline-treated SCA3 84Q vs. SCA3 15Q, 217.66 ± 11.78 μm vs. 257.15 ± 7.94 μm; *p* < 0.05); however, significant restoration was noted in the IGF-1-treated SCA3 84Q mice (IGF-1-treated SCA3 84Q vs. saline-treated SCA3 84Q, 246.61 ± 4.81 μm vs. 217.66 ± 11.78 μm; *p* < 0.05) (Figure 2c). The average thickness of the ML in the saline-treated SCA3 84Q mice was lower than that in the SCA3 15Q mice (119.79 ± 2.69 μm vs. 129.18 ± 2.69 μm; *p* > 0.05). However, the thickness of the ML was restored in the IGF-1-treated SCA3 84Q mice compared with the saline-treated SCA3 84Q mice (137.24 ± 4.07 μm vs. 119.79 ± 2.69 μm; *p* < 0.05) (Figure 2d).

### 3.3. IGF-1 Reduced Mutant Ataxin-3 Protein Expression in the SCA3 Mice

Our previous data indicated that ataxin-3 immunohistochemical (IHC) staining demonstrated strong immunoreactivity to the SCA3 84Q mice. Consistent with the previous findings, we observed that ataxin-3 expression was significantly increased in the PCs of the saline-treated SCA3 84Q mice. After the IGF-1 treatment, the immunoreactivity patterns of ataxin-3 were moderately reduced in the IGF-1-treated SCA3 84Q mice, while the protein expression level of ataxin-3 was significantly decreased. Although the SEM was higher in the saline-treated SCA3 84Q and IGF-1-treated SCA3 84Q groups, there was still a significant change comparison (Figure 3a,b). Toxicity of the mutant ataxin-3 protein may be attributable to misfolding and abnormal aggregation [4]. Therefore, protein aggregation detected using the ProteoStat^®^ dye colocalized with ataxin-3 was evaluated through immunofluorescence staining. Compared with the SCA3 15Q mice, the saline-treated SCA3 84Q mice demonstrated a significant increase in the aggregation signal in PCs. After the IGF-1 treatment, the mean intensity of the aggregation signal was significantly decreased and returned to the same level as that in the SCA3 15Q mice (Figure 3c).

### 3.4. IGF-1 Enhanced the Autophagy Pathway

To determine whether the IGF-1-caused reduction in the mutant ataxin-3 protein expression was related to autophagy, we measured the expression of autophagy-related proteins, namely Beclin1, p62, Atg7, and LC3-II, and the lysosomal marker Lamp2. We found that the expression levels of Beclin1 and LC3-II but not of p62, Atg7, and Lamp2 were significantly decreased in the saline-treated SCA3 84Q mice compared with the SCA3 15Q mice, indicating that the autophagy mechanism of the SCA3 84Q mice was impaired. However, the expression levels of Beclin1 and LC3-II were significantly increased after the IGF-1 treatment, indicating the possible restoration of the autophagy inducement (Figure 4).

### 3.5. IGF-1 Enhanced the Mitochondrial Function in the SCA3 Mouse Cerebellum

After euthanizing the mice, a part of their cerebellum was ground immediately, and O2k was used to detect mitochondrial respiration in the cerebellum. The mitochondrial function of the IGF-1-treated SCA3 84Q mice was more satisfactory than that of the saline-treated SCA3 84Q mice, with the IGF-1-treated SCA3 84Q mice having more OXPHOS, higher maximal mitochondrial phosphorylation respiration capacity (Max-Ox), and better electronic delivery system (Max-U) (Figure 5a). Formation of 8-hydroxy-2′-deoxyguanosine (8-OHdG) is a marker of mitochondrial DNA oxidative damage [31]. The arrangement of PCs could be clearly observed in IHC staining. The ImageJ software was used to determine the expression of the 8-OHdG protein in each PC. The cerebellar sections were selected from 4–5 mice in each group, and 10 microscopic fields were examined in each section accordingly. We observed that the 8-OHdG level was significantly increased in the PCs of the saline-treated SCA3 84Q mice. After the IGF-1 treatment, the 8-OHdG level was moderately reduced (Figure 5b). To investigate the effects of IGF-1 on mouse cerebellar mitochondria, we performed the Western blot analysis of the mitochondrial dynamics-related proteins. Dynamin-related protein 1 (Drp-1), optic atrophy protein 1 (Opa1), and mitofusin-2 (Mfn2) were used to detect mitochondrial dynamics. SCA3 exhibits more mitochondrial fission [32]. As shown in Figure 5c, we observed a slight increase in the fission protein phospho-Drp1 (p-Drp1) in the saline-treated SCA3 84Q mice but a significant decrease in the fusion protein Mfn2. However, the expression levels of these proteins were recovered after the IGF-1 treatment, indicating that mitochondria tended to undergo more fusion.

### 3.6. Nf-L Concentration in Plasma

Since an increase in Nf-L is associated with brain injury and atrophy [33], we evaluated the concentration of Nf-L in mouse plasma. A significant increase was observed in the saline-treated SCA3 84Q mice compared with the SCA3 15Q mice (*p* < 0.05). After the IGF-1 administration, the Nf-L concentration in the IGF-1-treated SCA3 84Q mice declined slightly, and there was no significant difference between the SCA3 15Q and IGF-1-treated SCA3 84Q mice (Figure 6).

### 3.7. Histopathological Findings in the Major Organs

GH might increase the occurrence of liver, lung, and kidney cancer. However, IGF-1 is the downstream promotor of GH. To ensure that the administration of IGF-1 is relatively safe, we performed H&E staining to determine whether the pathological sections of the liver, lung, and kidney were abnormal. As shown in Figure 7, irrespective of IGF-1 treatment, no obvious histopathological findings were noted.

## 4. Discussion

The results of the present study revealed that the IGF-1 treatment restored motor function, reduced neuronal cell death, and effectively prevented cerebellar atrophy in the SCA3 84Q transgenic mice. The mechanism of neuroprotection involves the enhancement of the autophagy process and the restoration of mitochondrial dynamics or oxidative phosphorylation after the IP injection of IGF-1. However, although the course of treatment was the same, IGF-1 was not as effective as GH [12]. This difference might be associated with the different levels of regulation between GH and IGF-1. In contrast to the downstream promotor IGF-1, GH directly supports growth and metabolism and increases muscle mass and strength [34].

IGF-1 inhibits cell apoptosis and stimulates cell proliferation to promote cancer development. Many epidemiological studies have reported that circulating IGF-1 is positively correlated with various primary cancers such as those of the breast, colon, and prostate [35]. IGF-1 induces the growth and metastasis of hepatocellular carcinoma by inhibiting protease-induced cathepsin B degradation [36]. Stimulation of the IGF-1 receptor (IGF-1R) could promote malignant transformation, cell proliferation and differentiation, and apoptosis inhibition [37]. However, systemic IGF-1 treatment did not alter tumor development in the mice injected with the naturally occurring densities of IGF-1R fibroblasts; this effect was dependent on the dose and the IGF-1R expression level [38]. Grimberg indicated that circulating IGF-1 may be related to the risk of different types of cancer but does not have a causal role in cancer formation; local changes in mechanisms controlling cell growth are required [39]. The findings of the present study indicated that treatment with IGF-1 for 9 months did not increase the circulating IGF-1 concentration; however, the expression levels of IGF-1 and IGF-1R were restored to those observed in the normal SCA3 15Q mice [40]. Furthermore, no carcinogenicity was noted in the pathological sections of the liver, lungs, and kidneys. These results indicated that the dosage of IGF-1 was safe (50 mg/kg) and did not interfere with IGF-1 metabolism and gene expression during the 9-month treatment in the SCA3 84Q mice.

Our previous study reported that behavioral functions were restored in the SCA3 84Q transgenic mice after 9 months of GH intervention, including those examined using the rotarod and locomotor tests [12]. This finding implied that after GH therapy, gait function was restored even if the disease was already progressing to a symptomatic stage in the SCA3 mice aged > 9 months. IGF-1, a downstream molecule, might exert some effect on the SCA3 mice model. Saenger et al. reported that polyethylene glycol-modified IGF-1 treatment in a SOD1-G93A ALS mouse with a low transgene copy number significantly delayed symptom onset, increased forelimb grip strength, and improved rotarod performance [41]. Eleftheriadou et al. used the αCAR-targeted vector to investigate the neuroprotective effects of IGF-1 on the SOD1-G93A mouse model. Their results indicated that αCAR IGF-1 LV treatment significantly prolonged the motor function in SOD1-G93A mice. The rotarod performance of the female mice treated with αCAR IGF-1 LV was more favorable than that of the control group. The findings of a footprint analysis showed that αCAR IGF-1 LV-treated mice had more satisfactory walking patterns [42]. Pristerà et al. reported that the distance traveled by the IGF-1 conditional KO mice was smaller than that traveled by the control mice, indicating a decrease in spontaneous locomotive activity [43]. In the present study, IGF-1 treatment for 9 months gradually improved the latency to fall, especially after the age of 17–18 months. In the behavioral and locomotor tests, the distance moved and the frequency of zone changes were more favorable. In the digital footprint analysis, the balance capability of the IGF-1-treated SCA3 84Q mice was similar to that of the SCA3 15Q control mice; however, this result was not statistically significant.

IGF-1, an anabolic and neuroprotective agent, promotes the survival of neurons by blocking apoptosis [44]. IGF-1 overexpression in transgenic mice increased the number of neurons in the GL and ML, the total number of neurons, and the total number of synapses in the ML [45]. The increased number of neurons in the dentate gyrus during the postnatal period not only prevented neuronal death, but also increased neurogenesis [46]. These results are consistent with those reported by Chrysis et al. [47]. IGF-1 is necessary for the survival of PCs in the neonatal cerebellum [44]. Disruption of PC viability alters the function of the whole cerebellum, leading to cerebellar ataxia [48]. The findings of the present study indicated that IGF-1 exerted neuroprotective effects by preserving PCs and increasing the thickness of the cerebellar cortex in the IGF-1-treated SCA3 84Q mice; this finding is consistent with that observed in GH-treated SCA3 84Q mice in our previous study [12]. IGF-1 exerts neuroprotective effects through mitochondrial activation [49]. Mitochondrial dysfunction is related to various neurodegenerative diseases, including polyQ diseases such as SBMA and HD [50]. In our SCA3 cell model, the basal respiratory rate and ATP-linked respiration and respiratory capacity were significantly decreased [51]. Liver IGF-1 knockout (LID) mice demonstrated a decline in OXPHOS capacity and OXPHOS coupling efficiency in hippocampal neurons [52]. IGF-1 might exert neuroprotective effects on SCA3 by supporting mitochondrial function accompanied with a significant increase in 8-OHdG, a biomarker of oxidative damage. Similar findings were observed in an IGF-1-treated nonalcoholic steatohepatitis mouse model [53]. Dynamic mitochondrial homeostasis plays a crucial role in neuron survival and age-related neurodegeneration [54,55]. The damaged mitochondria expressing abnormally increased fission in contrast to fusion are cleared through mitophagy, and the disruption of the mitophagy clearance of dysfunctional mitochondria induces cell death [56]. Drp1 is responsible for regulating fission, whereas Opa1 and Mfn1/2 regulate fusion. Shirendeb et al. reported that in HD, a polyQ disease, the inhibition of the interaction of mutant *Htt* with Drp1 reduced mitochondrial fission and increased mitochondrial fusion in neurons [57]. Furthermore, Ribeiro et al. reported that low doses of IGF-1 reduced Drp1 phosphorylation in HD striatal cells [58]. The dysfunction of cerebellar coordination and balance as well as early death might be observed in mice with the knockout of *Opa1* and *Mfn2* [59,60]. Thus, these findings are in accordance with our result that the cerebellar mitochondria of the IGF-1-treated SCA3 84Q mice underwent more fusion. Thus, the improvement of mitochondrial dynamics could be involved in the restoration of mitochondrial function and the reduction in the death of cerebellar neurons under IGF-1 treatment.

In our previous study, GH treatment exerted a neuroprotective effect and reduced ataxin-3 expression in SCA3 84Q mice [12]. In the present study, IGF-1, the downstream promotor of GH, reduced the immunoreactivity and protein expression of ataxin-3 in the cerebellum of the SCA3 84Q mice. Palazzolo et al. reported that IGF-1 could reduce androgen receptor (AR) aggregation and increase AR clearance in cells expressing AR65Q, a cell model of SBMA, a polyQ neurodegenerative disease. Simultaneously, the overexpression of IGF-1 in the muscles of SBMA mice prolonged their lifespan and rescued behavioral abnormalities [61]. Regulation of the autophagic flux might ameliorate the progression of neurodegenerative diseases including SCA3 [30,62]. In the animal models of SCA3 and HD, the dysregulation of autophagy caused the accumulation of toxic mutant proteins [62,63]. Autophagy can clear aggregated misfolding proteins by inducing the expression of autophagy-related proteins including Beclin1, p62, and LC3. IGF-1 induces autophagy by upregulating the Beclin1 and LC3-II levels [64]. Wen et al. reported that the intramuscular injection of human IGF-1 with a self-complementary adeno-associated virus vector in the cells and mouse models of ALS could upregulate mitochondrial autophagy and suppress mitochondrial apoptosis [22]. Nevertheless, Beclin1 and p62 played different roles in the pathogenesis of autophagy. Beclin1 induces autophagosome formation while p62 acts as an adaptor protein and delivers ubiquitinated substrates to the proteasome [65]. More specifically, overexpression of Beclin1 has previously been reported to be associated with neuroprotection [66]. In this study, the IGF-1-treated SCA3 84Q mice exhibited increased autophagy, as indicated by the upregulation of Beclin1 and LC3-II. Thus, IGF-1-related autophagy might be a mechanism through which mutant proteins are cleared and cerebellar degeneration is prevented. However, the detailed correlation between IGF-1 and autophagy flux using autophagy inhibitors in SCA3 mice remains to be investigated.

In addition, in recent years, researchers have been working to find biomarkers for SCAs. Neurofilaments are the main components of the neuron cytoskeleton protein, composed of three subunits called Nf-L, neurofilament medium chain, and neurofilament heavy chain [67]. There is a lot of evidence that the level of Nf-L in the blood can be used as a biomarker for neurodegenerative diseases. Nf-L levels also reflect disease severity, longitudinal disease progression, CAG repeat length and age [68]. In other words, the Nf-L levels increase with proximity to the estimated onset [69]. Therefore, Nf-L is a potential candidate biomarker for therapeutic response. Indeed, this study showed a significant increase in the saline-treated SCA3 84Q mice compared with the SCA3 15Q mice. The Nf-L concentration in the IGF-1-treated SCA3 84Q mice declined slightly after the IGF-1 treatment, and there was no significant difference compared with the SCA3 15Q mice.

In summary, the findings of the present study revealed that IGF-1 ameliorated the impairment of motor functions, suppressed the degeneration of PCs, preserved the thickness of ML and GL, enhanced the function of mitochondria, and reduced oxidative stress and the mutant ataxin-3 protein expression including protein aggregate formation in the cerebellum. IGF-1 was not found to be carcinogenic in the SCA3 84Q mice at the dose and time used in the present study. Considering the benefits of early treatment [41], treatment efficacy can be improved in SCA3 mice before 7.5 months of age because the balance and coordination ability of SCA3 84Q mice start to decline at the age of 7.5–13 months and their limb position starts becoming abnormal at the age of 6 months [70]. On the basis of the results of this study, future studies should investigate the most appropriate treatment approaches involving the use of IGF-1 and validate its in vivo efficacy for SCA3 treatment.

## Figures and Tables

**Figure 1 biomedicines-10-00505-f001:**
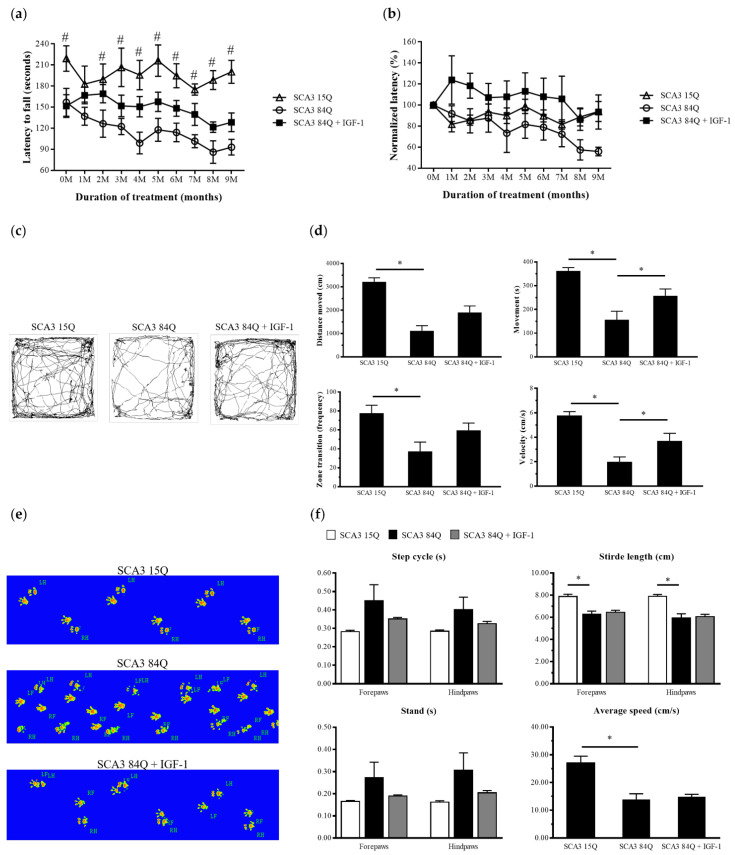
IGF-1 prevented impairment of the motor function in the SCA3 mice. (**a**) Latency to fall (time in seconds for which the mice persisted on the rotarod) for the SCA3 15Q mice and the saline- and IGF-1-treated SCA3 84Q mice during the 9 months of treatment. (**b**) Within the same group, the latency to fall at pretreatment was normalized to 100%. (**c**) EthoVision XT 7.0 software was used to analyze trajectories of the mice in the behavioral test. (**d**) The distance of movement, time of movement, frequency of zone change, and average velocity were included in transformed indices. (**e**) Captured images of the single stance for each paw. (**f**) Catwalk parameters included the step cycle, stride length, stand, and average speed. The data are presented as the means ± SEM. Note: # *p* < 0.05 denotes statistical significance in the saline-treated SCA3 84Q mice compared with the SCA3 15Q mice; * *p* < 0.05 indicates a significant difference.

**Figure 2 biomedicines-10-00505-f002:**
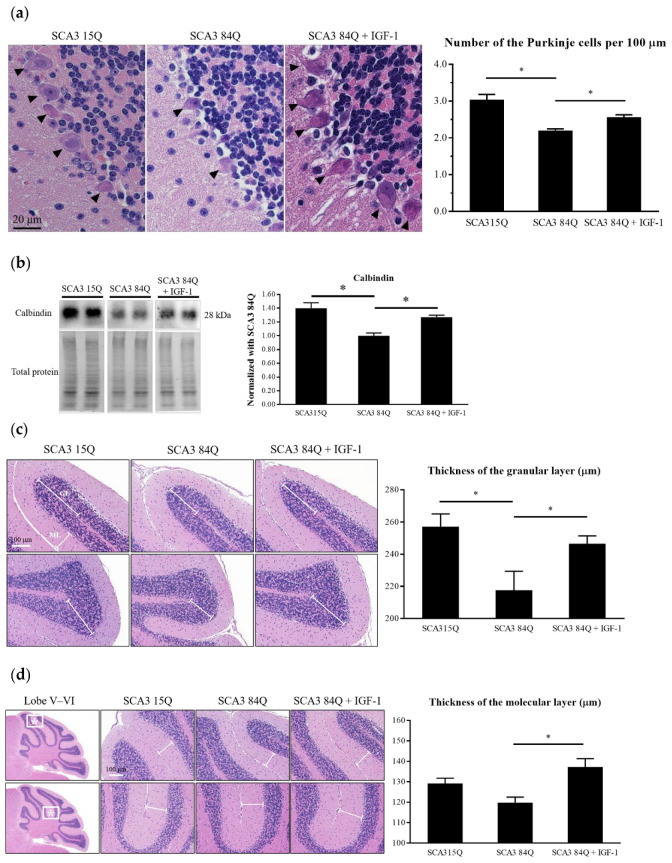
IGF-1 prevented the death of the PCs in the cerebellum of the SCA3 84Q mice. (**a**) The arrows indicate the PCs located at the PCL (right panel). The average number of the PCs per 100 μm in the posterior lobules of the cerebellum is presented in the bar graph (mean ± SEM) (left panel). SCA3 15Q, n = 6; SCA3 84Q, n = 8; SCA3 84Q + IGF-1, n = 8. (**b**) Western blot analysis of calbindin (left panel). Relative expression levels of calbindin in the cerebellum (mean ± SEM) (right panel). SCA3 15Q, n = 4; SCA3 84Q, n = 4; SCA3 84Q + IGF-1, n = 4. (**c**) The lines indicate the distance from the tip of the granular layer (GL) to the white matter (right panel). Histogram showing the thickness of the GL (mean ± SEM) (left panel). SCA3 15Q, n = 5; SCA3 84Q, n = 6; SCA3 84Q + IGF-1, n = 8. (**d**) The frames are the sampling area of the Figure and the lines refer to the distance from the PCL to the edge of the molecular layer (ML) (right panel). Histogram showing the thickness of the ML (mean ± SEM) (left panel). SCA3 15Q, n = 5; SCA3 84Q, n = 6; SCA3 84Q + IGF-1, n = 8. Note: * *p* < 0.05 indicates a significant difference.

**Figure 3 biomedicines-10-00505-f003:**
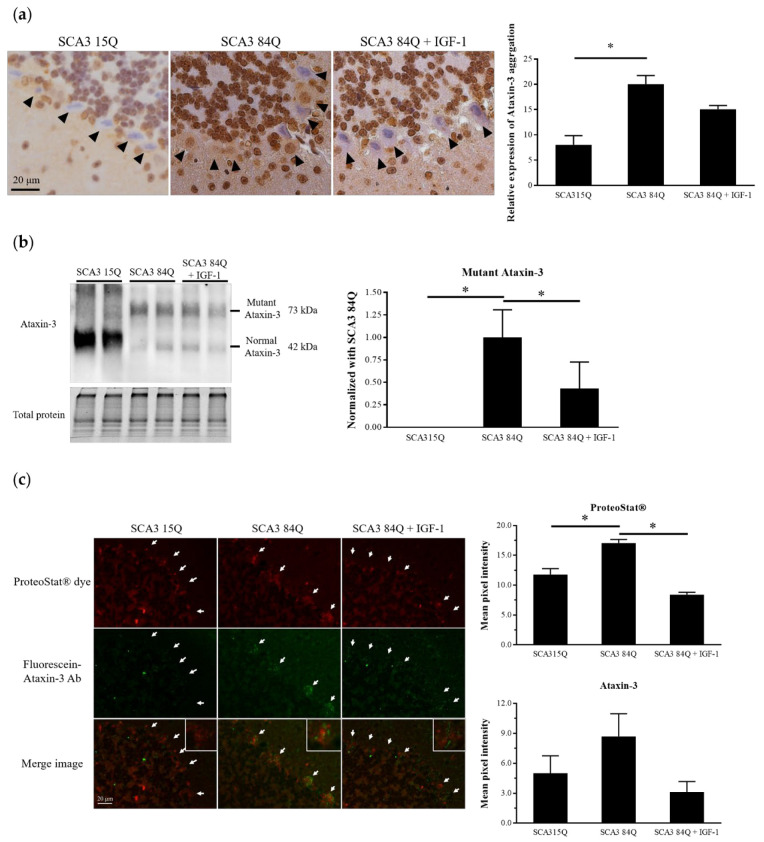
IGF-1 reduced the ataxin-3 protein level in the cerebellum of the SCA3 84Q mice. (**a**) Immunochemical staining of ataxin-3 in the cerebellum. The black arrows indicate PCs (right panel). Histograms show the means ± SEM (left panel). SCA3 15Q, n = 4; SCA3 84Q, n = 4; SCA3 84Q + IGF-1, n = 5. (**b**) Western blot confirming ataxin-3 expression in the mouse cerebellum (left panel). Quantification of the ataxin-3 level relative to the total protein level (mean ± SEM) (right panel). SCA3 15Q, n = 4; SCA3 84Q, n = 4; SCA3 84Q + IGF-1, n = 5. (**c**) Slices of the cerebellum of two mice in each group were selected and double-labeled using an aggresome detection kit (red) and an Alexa 488-conjugated secondary IgG against the anti-ataxin-3 antibody (green), and fluorescence intensities of 30–40 PCs in each mouse were examined using the ImageJ software. The white arrows indicate PCs. SCA3 15Q, n = 2; SCA3 84Q, n = 2; SCA3 84Q + IGF-1, n = 2. Note: * *p* < 0.05 indicates a significant difference.

**Figure 4 biomedicines-10-00505-f004:**
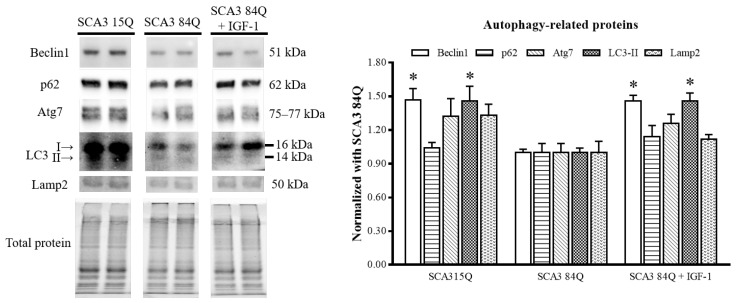
Expression of the autophagic influx in the SCA3 mice. Representative Western blots of the autophagy-related markers (right panel). Quantitative results of the autophagy-related proteins were normalized to those of total protein (mean ± SEM) (left panel). SCA3 15Q, n = 4; SCA3 84Q, n = 4; SCA3 84Q + IGF-1, n = 4. Note: * *p* < 0.05 indicates a significant difference.

**Figure 5 biomedicines-10-00505-f005:**
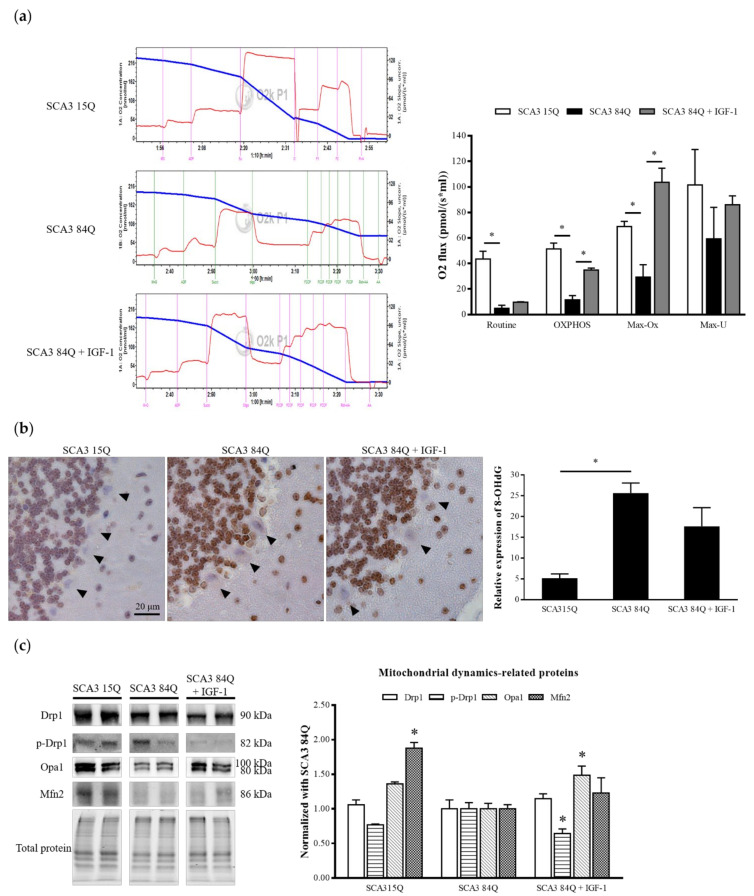
Expression of the mitochondrial function in the SCA3 mice. (**a**) Typical trace of respirometry measurements recorded using an Oroboros O2k with 2 mg/mL of the cerebellum. The blue curve indicates the oxygen concentration in the sealed chamber, whereas the red curve shows the oxygen consumption of tissue cells (left panel). Oxygen consumption of cells at different mitochondrial stages was corrected for ROX, and the respiratory capacities in the routine, OXPHOS, Max-Ox, and Max-U states were plotted as the means ± SEM (right panel). SCA3 15Q, n = 3; SCA3 84Q, n = 3; SCA3 84Q + IGF-1, n = 2. (**b**) The 8-OHdG protein expression in the cerebellum sections by IHC staining analysis; the arrows indicate PCs (left panel). Histogram shows the mean ± SEM (right panel). SCA3 15Q, n = 4; SCA3 84Q, n = 5; SCA3 84Q + IGF-1, n = 5. (**c**) Western blot was performed to analyze the expression of mitochondrial dynamics-related proteins (right panel). Quantification of mitochondrial dynamics-related proteins (left panel). SCA3 15Q, n = 4; SCA3 84Q, n = 4; SCA3 84Q + IGF-1, n = 4. Note: * *p* < 0.05 indicates a significant difference.

**Figure 6 biomedicines-10-00505-f006:**
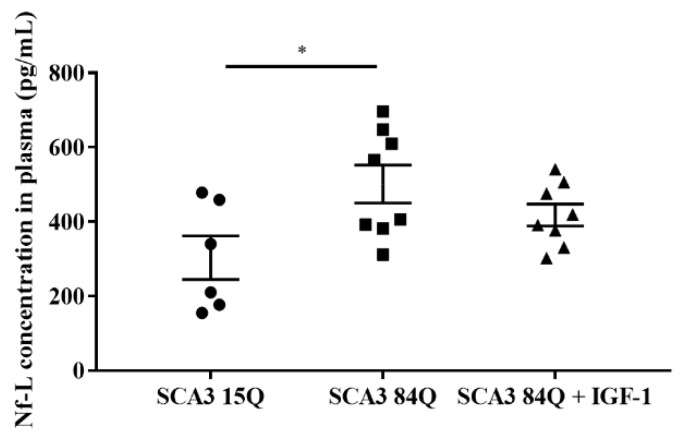
Plasma concentration of Nf-L. SCA3 15Q, n = 6; SCA3 84Q, n = 8; SCA3 84Q + IGF-1, n = 8. Note: * *p* < 0.05 was considered a statistically significant difference.

**Figure 7 biomedicines-10-00505-f007:**
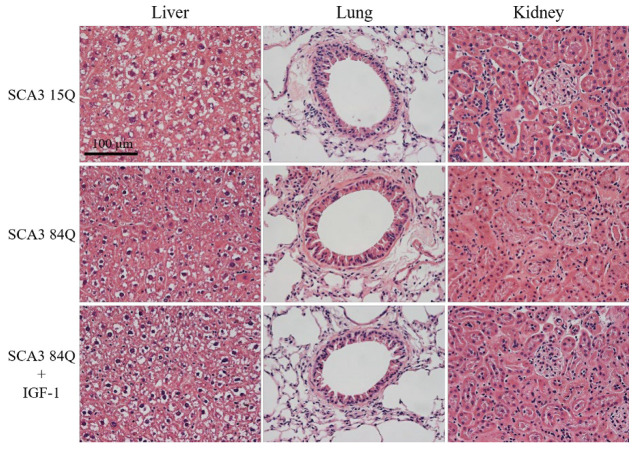
Tissue sections of the liver, lung, and kidney. No significant histopathological findings of the kidneys, liver, and lungs were observed in the SCA3 15Q, saline-treated SCA3 84Q, and IGF-1-treated SCA3 84Q mice. n = 3 in all the groups.

## Data Availability

The data presented in this study are available on request from the corresponding author.

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
