# Peer review of "IGF-1 as a Potential Therapy for Spinocerebellar Ataxia Type 3"

_biomedicines, 2022, doi:10.3390/biomedicines10020505_

Round 1
Reviewer 1 Report
The Research article by Yong Shiou Lin et al., entitled “IGF-1 as a Potential Therapy for Spinocerebellar Ataxia Type 3” investigated the efficacy of IGF-1 in vivo for SCA3 treatment.
SCA3 84Q transgenic mice were administered a weekly IP injection of IGF-1 for 9 months. The results of the present study revealed that IGF-1 treatment restored motor function, reduced neuronal cell death, and effectively prevented cerebellar atrophy in the SCA3 84Q transgenic mice. Indeed, compared with control mice SCA3 15Q, the SCA3 84Q mice treated with IGF-1 exhibited the significant restoration of locomotor function and minimized degeneration of the cerebellar cortex, as indicated by the survival of more Purkinje cells with more favorable mitochondrial function along with a decrease in oxidative stress caused by DNA damage.
Although it is well written and expanded there are some points to review:
- Figure 3B. (b) Western blot confirming ataxin-3 expression in the mouse cerebellum (right panel). Quantification of the ataxin-3 level relative to the total protein level (mean ± SEM; left panel).
Please, correct the caption reversing left with right. Moreover, I would just point the attention of the authors on the Ataxin 3 western blot (Figure 3B + Original images file): quantification was performed on two different experiments, right? In the Supplementary file are showed 2 different blots, one of which is reported twice at a different magnification, so in total we have 3 images but for only 2 assays. In the densitometry graph of Mutant Ataxin 3, the SEM for SCA3 84Q and the SEM for SCA3 84Q+IGF-1 seem to overlap, however, the difference in Mutant Ataxin 3 expression is significant. Don't the authors have a third replicated of the experiment to add, to try to reduce the SEM? If not, the difference, albeit significant, is slight. I suggest the authors to underline this evidence in the main text.
- Figure 3C. Please, improve images resolution. I would suggest to the authors to create some enlargements on the co-localization points (i.e. for the saline-treated SCA3 84Q merge image), to show the yellow dots clearly.
- Line 335-338. “Expression levels of Beclin1 and LC3-II but not p62, Atg7, and Lamp2 were significantly decreased in the saline-treated SCA3 84Q mice compared with the SCA3 15Q mice, indicating that the autophagy mechanism of the SCA3 84Q mice was impaired. However, the expression levels of Beclin1 and LC3-II were significantly increased after IGF-1 treatment, indicating the possible restoration of the autophagy flux (Fig. 4).” Rather than “restoration of the autophagy flux” I would suggest that the authors remain cautious by simply talking about the “restoration of the autophagy process”. LC3 II and Beclin I are autophagy-related markers in the first steps of autophagy activation/induction (=autophagosome formation). To better investigate the autophagy flux, the authors should have compared the expression ratio LC3II/LC3I with the expression of p62 in SCA3 84Q mice compared with SCA3 15Q and SCA3 84Q+IGF-1 mice, even better if in the presence or absence of autophagy inhibitors (i.e. 3MA, Bafilomycin A1).
- Line 517, Discussion paragraph. The authors declare: “IGF-1 was not found to be carcinogenic in the SCA3 84Q mice at the dose used in the present study”. Better to add “…at the dose and time used…”.
Author Response
Thank you for providing us the opportunity to submit a revised manuscript. We have checked the comments carefully and revised the manuscript accordingly and used the “Track Changes” function. A complete, point-by-point response to the reviewer critiques follows.
Thank you for providing us the opportunity to submit a revised manuscript. We have checked the comments carefully and revised the manuscript accordingly and used the “Track Changes” function. A complete, point-by-point response to the reviewer critiques follows.
Response to Reviewer 1 Comments
The Research article by Yong Shiou Lin et al., entitled “IGF-1 as a Potential Therapy for Spinocerebellar Ataxia Type 3” investigated the efficacy of IGF-1 in vivo for SCA3 treatment.
SCA3 84Q transgenic mice were administered a weekly IP injection of IGF-1 for 9 months. The results of the present study revealed that IGF-1 treatment restored motor function, reduced neuronal cell death, and effectively prevented cerebellar atrophy in the SCA3 84Q transgenic mice. Indeed, compared with control mice SCA3 15Q, the SCA3 84Q mice treated with IGF-1 exhibited the significant restoration of locomotor function and minimized degeneration of the cerebellar cortex, as indicated by the survival of more Purkinje cells with more favorable mitochondrial function along with a decrease in oxidative stress caused by DNA damage.
Although it is well written and expanded there are some points to review:
- Figure 3B. (b) Western blot confirming ataxin-3 expression in the mouse cerebellum (right panel). Quantification of the ataxin-3 level relative to the total protein level (mean ± SEM; left panel).
Please, correct the caption reversing left with right. Moreover, I would just point the attention of the authors on the Ataxin 3 western blot (Figure 3B + Original images file): quantification was performed on two different experiments, right? In the Supplementary file are showed 2 different blots, one of which is reported twice at a different magnification, so in total we have 3 images but for only 2 assays. In the densitometry graph of Mutant Ataxin 3, the SEM for SCA3 84Q and the SEM for SCA3 84Q+IGF-1 seem to overlap, however, the difference in Mutant Ataxin 3 expression is significant. Don't the authors have a third replicated of the experiment to add, to try to reduce the SEM? If not, the difference, albeit significant, is slight. I suggest the authors to underline this evidence in the main text.
> Thank you for pointing this out. We have corrected the text to reverse left with right in lines 335 and 336. According to our Supplementary file, indeed, the quantification was performed on two different experiments. As the reviewer mentioned one of Ataxin 3 western blot is reported twice at a different magnification is because the below one is put in the article and the editor asked us to attach the original, unedited image. So, the upper right image in Supplementary file page 2 is an unedited image, but we also refer to our number of samples in line 338 in the manuscript (SCA3 15Q, n = 4; SCA3 84Q, n = 4; SCA3 84Q+IGF-1, n = 5.) According to the reviewer’s suggestion, we added the description about the significant change in Mutant Ataxin 3 in SCA3 84Q and SCA3+IGF-1 groups (line 323-324).
- Figure 3C. Please, improve images resolution. I would suggest to the authors to create some enlargements on the co-localization points (i.e. for the saline-treated SCA3 84Q merge image), to show the yellow dots clearly.
> Thank you for the comments. We have created enlargements for the merge images in Figure 3c.
- Line 335-338. “Expression levels of Beclin1 and LC3-II but not p62, Atg7, and Lamp2 were significantly decreased in the saline-treated SCA3 84Q mice compared with the SCA3 15Q mice, indicating that the autophagy mechanism of the SCA3 84Q mice was impaired. However, the expression levels of Beclin1 and LC3-II were significantly increased after IGF-1 treatment, indicating the possible restoration of the autophagy flux (Fig. 4).” Rather than “restoration of the autophagy flux” I would suggest that the authors remain cautious by simply talking about the “restoration of the autophagy process”. LC3 II and Beclin I are autophagy-related markers in the first steps of autophagy activation/induction (=autophagosome formation). To better investigate the autophagy flux, the authors should have compared the expression ratio LC3II/LC3I with the expression of p62 in SCA3 84Q mice compared with SCA3 15Q and SCA3 84Q+IGF-1 mice, even better if in the presence or absence of autophagy inhibitors (i.e. 3MA, Bafilomycin A1).
> We thank the reviewer for the constructive comments. According to Han et al., the authors had proved IGF-1 could induce autophagy in porcine primary granulosa cells by increasing Beclin I and LC3 II (line 505-507, ref. 64). Therefore, in this study, the observation of upregulation of the expression levels of Beclin1 and LC3-II after IGF-1 treatment, we concluded that autophagy flux might be restored. However, we didn’t use the autophagy inhibitor in mice to disprove; thus we have modified the text in the result (line 351) and discussion (line 510-514 and 517-518) to include detailed descriptions.
- Line 517, Discussion paragraph. The authors declare: “IGF-1 was not found to be carcinogenic in the SCA3 84Q mice at the dose used in the present study”. Better to add “…at the dose and time used…”.
> Thank you for the comments. We have modified the text in the discussion (line 534-535).

Reviewer 2 Report
This study investigated the efficacy of IGF-1, a downstream mediator of 20GH, in vivo for SCA3 treatment. The authors use standard mouse models for this pathology and standard experimental metrics of assessment like rotarod, open field test, catwalk gait analysis, histology & immunochemistry, and mitochondrial function assessment.
Thee study is well presented, the experimental design appropriate, and the results and discussion well described. The primary issue is the lack of details and apparent rigor of the statistical analysis.
MAJOR
The statistical analysis section of the methods only reports the use of a student's t-test in Microsoft Excel to compare significant differences. However, there is no mention of a Bonferroni or other similar correction when doing multiple comparisons. Without correcting the p-value threshold for significance for multiple comparisons, the chance of a Type I error (a false positive) is higher and the results are technically misrepresented. This issue applies for pretty much every bar graph shown in the manuscript. This needs corrected in the Methods description, figure captions, and results.
The temporal assessment of rotarod as a function of treatment duration is nice in Figure 1 a,b. However, I would recommend performing a log-rank test or a similar analysis to determine statistical significance for the entire trend line. This would not be appropriate of comparing differences between each time point, but it does give a nice wholistic analysis of the entire treatment timeline.
MINOR
Finally, the multiple features measured experimentally is a major asset of this work. It would be nice to have an integrated analysis to assess relative impact of the treatment on each feature. This would not be required, but it would add to the work -- perhaps something like a standardized aggregate effect size analysis on each experimental feature measured.
Author Response
Thank you for providing us the opportunity to submit a revised manuscript. We have checked the comments carefully and revised the manuscript accordingly and used the “Track Changes” function. A complete, point-by-point response to the reviewer critiques follows.
Thank you for providing us the opportunity to submit a revised manuscript. We have checked the comments carefully and revised the manuscript accordingly and used the “Track Changes” function. A complete, point-by-point response to the reviewer critiques follows.
Response to Reviewer 2 Comments
This study investigated the efficacy of IGF-1, a downstream mediator of 20GH, in vivo for SCA3 treatment. The authors use standard mouse models for this pathology and standard experimental metrics of assessment like rotarod, open field test, catwalk gait analysis, histology & immunochemistry, and mitochondrial function assessment.
The study is well presented, the experimental design appropriate, and the results and discussion well described. The primary issue is the lack of details and apparent rigor of the statistical analysis.
MAJOR
The statistical analysis section of the methods only reports the use of a student's t-test in Microsoft Excel to compare significant differences. However, there is no mention of a Bonferroni or other similar correction when doing multiple comparisons. Without correcting the p-value threshold for significance for multiple comparisons, the chance of a Type I error (a false positive) is higher and the results are technically misrepresented. This issue applies for pretty much every bar graph shown in the manuscript. This needs corrected in the Methods description, figure captions, and results.
> Thank you for the comments. According to the reviewer’s comments, statistical analysis was determined by one-way analysis of variance (ANOVA) followed by the Bonferroni multiple-comparison post hoc test or two-way repeated-measures ANOVA followed by the Bonferroni multiple-comparison post hoc test (Rotarod test). We have modified the text and statistical analysis methods. Data are shown as the mean ± S.E.M and statistical significance was set at P < 0.05. (line 229-236).
The temporal assessment of rotarod as a function of treatment duration is nice in Figure 1 a,b. However, I would recommend performing a log-rank test or a similar analysis to determine statistical significance for the entire trend line. This would not be appropriate of comparing differences between each time point, but it does give a nice wholistic analysis of the entire treatment timeline.
> Thank you for the comments. According to the reviewer’s recommendations, statistical analysis was determined by two-way repeated-measures ANOVA followed by the Bonferroni multiple-comparison post hoc test (Rotarod test). We have modified the text and statistical analysis methods. (line 229-236).
MINOR
Finally, the multiple features measured experimentally is a major asset of this work. It would be nice to have an integrated analysis to assess relative impact of the treatment on each feature. This would not be required, but it would add to the work -- perhaps something like a standardized aggregate effect size analysis on each experimental feature measured.
> Thank you for your suggestion and we fully agree. We will pay attention to analysis and collection of data in the further work.